# *Angiotensin–Converting Enzyme (ACE) 1* Gene Polymorphism and Phenotypic Expression of COVID-19 Symptoms

**DOI:** 10.3390/genes12101572

**Published:** 2021-10-01

**Authors:** Naoki Yamamoto, Nao Nishida, Rain Yamamoto, Takashi Gojobori, Kunitada Shimotohno, Masashi Mizokami, Yasuo Ariumi

**Affiliations:** 1Genome Medical Sciences Project, National Center for Global Health and Medicine, Ichikawa 272-8516, Japan; lb-naonishida@hospk.ncgm.go.jp (N.N.); lbshimotohno@hospk.ncgm.go.jp (K.S.); mmizokami@hospk.ncgm.go.jp (M.M.); 2Intelligence for Medical and Nutritional Research, Tokyo 145-0065, Japan; yamamoto.rainy@gmail.com; 3Computational Bioscience Research Center, Biological and Environmental Sciences and Engineering, King Abdullah University of Science and Technology, Thuwal 23955, Saudi Arabia; takashi.gojobori@kaust.edu.sa; 4Division of Retroelement, Joint Research Center for Human Retrovirus Infection, Kumamoto University, Kumamoto 860-0811, Japan; ariumi@kumamoto-u.ac.jp

**Keywords:** SARS-CoV-2, COVID-19 symptoms, *ACE1* I/D polymorphism, Alu, RAAS, comorbidities

## Abstract

The renin–angiotensin–aldosterone system (RAAS) appears to play an important role in SARS-CoV-2 infection. Polymorphisms within the genes that control this enzymatic system are candidates for elucidating the pathogenesis of COVID-19, since COVID-19 is not only a pulmonary disease but also affects many organs and systems throughout the body in multiple ways. Most striking is the fact that ACE2, one of the major components of the RAAS, is a prerequisite for SARS-COV-2 infection. Recently, we and other groups reported an association between a polymorphism of the *ACE1* gene (a homolog of *ACE2*) and the phenotypic expression of COVID-19, particularly in its severity. The ethnic difference in *ACE1* insertion (I)/deletion (D) polymorphism seems to explain the apparent difference in mortality between the West and East Asia. The purpose of this review was to further evaluate the evidence linking *ACE1* polymorphisms to COVID-19. We searched the Medline database (2019–2021) for reference citations of relevant articles and selected studies on the clinical outcome of COVID-19 related to *ACE1* I/D polymorphism. Although the numbers of patients are not large enough yet, most available evidence supports the notion that the DD genotype adversely influences COVID-19 symptoms. Surprisingly, small studies conducted in several countries yielded opposite results, suggesting that the *ACE1* II genotype is a risk factor. This contradictory result may be the case in certain geographic areas, especially in subgroups of patients. It may also be due to interactions with other genes or to yet unexplained biochemical mechanisms. According to our hypothesis, such candidates are genes that are functionally involved in the pathophysiology of COVID-19, can act in concert with the *ACE1* DD genotype, and that show differences in their frequency between the West and East Asia. For this, we conducted research focusing on Alu-related genes. The current study on the *ACE1* genotype will provide potentially new clues to the pathogenesis, treatment, and diagnosis of SARS-CoV-2 infections.

## 1. Introduction

The novel coronavirus SARS-CoV-2, which is responsible for the current acute respiratory syndrome, COVID-19, belongs to the genus *Betacoronavirus* [1,2]. COVID-19 presents a high rate of morbidity and relatively elevated mortality. The emergence of COVID-19, along with the Spanish flu, is considered one of the most important threats to humankind in the last several centuries. In March 2020, the World Health Organization (WHO), which confirmed the spread and severity of SARS-CoV-2, declared that the COVID-19 outbreak recorded the previous month was a pandemic [3]. The infection has spread worldwide and currently affects more than 200 countries. As of early September 2021, the number of infected people has already exceeded 223 million, and more than 4.6 million have died [4]. Several countries, such as most European countries, the United States of America, India, Russia, Mexico, and South American countries, including Brazil and Peru, have been severely affected. Conversely, the number of cases and deaths in other regions, especially in largely populated East Asia, is much lower, and the racial differences are noteworthy [5]. Although more than 200 million people have fully recovered, many of the infected are still in crisis, requiring respiratory assistance, and suffering from sequelae.

SARS-CoV-2 enters host cells through the interaction of its spike protein with the entry receptor, ACE2 [6]. Accordingly, many studies targeting ACE2 have been conducted, including in the development of therapeutics [6,7]. The viral invasion process requires the priming of its spike proteins by TMPRSS2, a cell-surface serine protease [6] that is reported to be upregulated by androgens [8]. The binding of SARS-CoV-2 to ACE2 and processing of the S protein by TMPRSS2 may result in the downregulation and depletion of ACE2 [6,9]. Thus, the general mechanism of the multi-organ failure observed in COVID-19 may be associated with the systemic ACE2 expression that is ubiquitous throughout the body. As discussed below, ACE1 regulates ACE2 expression and activity through regulation of angiotensin (Ang) II levels. Since SARS-COV-2 infection downregulates ACE2 expression, reduced ACE2 expression may prevent viral infections. However, it also reduces the beneficial effects of ACE2 in the lungs and other organs. The protective RAAS has numerous beneficial actions, including anti-inflammatory, anticoagulative, and antifibrotic effects, along with endothelial and neural protection. Thus, the imbalance of the ACE1/ACE2 arms and the disruption of the homeostasis of the RAAS appear to be most significant at the onset of COVID-19 [1,10]. 

In the early stages of SARS-CoV-2 infection, common cold-like symptoms are often experienced, with low to moderate fever, dry cough, and malaise. However, as the condition worsened, it is accompanied by not only respiratory damage but also extra-pulmonary multi-organ failure, such as cardiovascular complications [10,11]. Severe COVID-19 is frequently associated with coagulation/fibrinolytic abnormalities characterized by increases in procoagulant factor levels, including fibrinogen, and increases in D-dimers that have been associated with higher mortality [12]. When blood vessels are clogged by blood clots, not only the lungs, but also various parts such as the heart and brain are damaged, and in the worst case, death may occur. Relatively high expression of ACE2 in the cardiovascular system, especially in vascular endothelial cells, may contribute to the predominant expression of symptoms such as thrombo-inflammation and hypercoagulative states in COVID-19 [13]. The incidence of venous thromboembolic events in Asian populations is significantly low. Incidence among the overall Asian population was estimated to be as low as 15–20% of the levels recorded in the Western world, although it has been increasing over time [14]. Mortality in COVID-19 patients has also been linked to the presence of a cytokine storm induced by viral infection [15], which also occurs in other viral and bacterial infections, such as MERS-CoV, influenza, and sepsis. Under this condition, excessive production of proinflammatory cytokines leads to the aggravation of lung pathology and widespread tissue damage, resulting in multi-organ failure and death. T-cell counts, including CD4 and CD8 cells, are significantly reduced in COVID-19 patients. T-cell numbers were negatively correlated to serum interleukin (IL)-6, IL-10, and tumor necrosis factor-α (TNF-α) concentration, and the surviving T cells appear functionally exhausted [16].

In the progression of COVID-19 to a severe or critical stage, various vulnerabilities and risk factors are recognized: old age, male gender, underlying medical conditions, and comorbidities such as hypertension; diabetes; obesity; chronic lung diseases; heart, liver, and kidney diseases; tumors; immunodeficiencies; and pregnancy [17]. The lifestyle of civilized society is deeply involved in most of these pathological conditions. People with comorbidities identified as risk factors for COVID-19 appear to have certain genetic vulnerabilities from the beginning. A common feature of these people is chronic low-grade inflammation (LGI) [18,19]. LGI is characterized by a chronic increase in inflammatory cytokines, such as IL-6, TNF-α, and IL-1 β. The excessive release of these proinflammatory mediators may be due to the accumulation of macrophages in the adipose tissue of obese individuals [20]. These people are likely to be more susceptible to SARS-CoV-2 infection and its exacerbations. Since many of these comorbidities are also reported to be closely associated with *ACE1* I/D polymorphisms, we proposed previously that these individuals may have a mild ACE1/ACE2 imbalance before viral infection, which increases their risk for developing severe cases of COVID-19 [18]. Similar to many other infections, COVID-19 patients have different clinical symptoms and host reactions, including those related to both gender and ethnicity. The male case fatality rate is noticeably higher than in females in 37 of the 38 regions, with an average male fatality rate 1.7 times greater than the average female rate [21]. We believe that understanding all of these risk factors, separately or in combination, can help unravel the mystery of COVID-19 aggravation.

## 2. Physiological and Pathological Roles of the RAAS

Most importantly, the plasma levels of Ang II from SARS-CoV-2-infected patients were significantly elevated when compared to that of healthy individuals. Moreover, the level of Ang II in SARS-CoV-2-infected patients was strongly associated with viral load and lung injury [22], and viral load was associated with increased disease severity and mortality [23]. All these data suggest that the imbalanced RAAS in patients may be caused by SARS-CoV-2. 

Lack of water and salt leads to the secretion of renin from the juxtaglomerular apparatus of the kidney, triggered by decreased renal perfusion pressure, sympathetic nerve excitement, and decreased blood Na levels. The protease renin acts on angiotensinogen produced by the liver in the blood to promote the conversion of a small molecule, Ang I. This hormone activates the adrenal hormone, aldosterone, to constrict blood vessels and promote salt absorption. Ang I is enzymatically converted to Ang II by ACE1 on the vascular endothelial cell membrane [24,25]. Ang II has a strong vasoconstrictive effect, promotes the production and secretion of aldosterone from the adrenal glands, and activates the secretion of vasopressin from the pituitary gland. Thus, these hormones have the important physiological significance of increasing blood pressure by constricting blood vessels and increasing the absorption of salt and water. 

When the degradation of Ang II by ACE2 is not appropriate, AT1 receptors (AT1Rs) are overstimulated. Heightened stimulation of AT1Rs can cause deleterious effects such as endothelial cell damage (vasculopathy), thrombosis (coagulopathy), and multisystem inflammation [26]. The counterregulatory and protective effects of ACE2 are consequently compromised through a lack of production of the protective peptides Ang-(1–7) and Ang-(1–9) and result in decreased stimulation of the Mas receptor (MasR). In contrast, when the blood circulation volume increases and the blood pressure rises, the secretion of renin is suppressed, and the function of this system is reduced. In this way, the RAAS forms a blood-circulation regulation mechanism by maintaining blood volume and increasing blood pressure. Since Ang II stimulates the production of inflammatory cytokines and chemokines, new potential uses for RAAS blockers have been suggested not only for hypertension or autoimmune diseases but also for COVID-19 [27].

The RAAS is an essential system for maintaining normal physical activity and is deeply involved not only in the cardiovascular and renal system but also in homeostasis of the whole body. High levels of Ang II, the major product of ACE1, result in various signal transductions because Ang II activates protein kinase C (PKC) and activator protein 1 (AP-1) and is involved in the production of free radicals such as reactive oxygen species (ROS) and the regulation of cell synthesis of several molecules, such as cytokines, chemokines, and transcription factors [28]. Activation of AT1R by ACE1/Ang II induces release of pro-inflammatory cytokines and chemokines, and causes cellular senescence, inflammation, and development of autoimmune dysfunction [29,30]. PKC increases ADAM17 enzyme activity of ADAM17, a pleiotropic enzyme. When ADAM17 is activated, many membrane proteins such as IL-6R and ACE2 are cleaved. As a result, activation of proinflammatory substances such as IL-6, signal transducer and activator of transcription (STAT)3, EGF receptor and TNF-α leads to cytokine storms [15,18,28]. 

## 3. Roles of ACEs and Ang II in the Immune System

In addition to regulating blood pressure, vasoconstriction, sodium intake, and potassium excretion, the RAAS is closely involved in immune function. RAAS imbalance is associated with the most common diseases of civilization, such as hypertension, atherosclerosis, cardiovascular diseases, diabetes, renal diseases, and several other degenerative diseases. Innate and adaptive immune responses participate in end-organ damage, and many of these pathological processes are attributed to the excessive activation of tissue, exerted especially by Ang II [30]. 

Different subpopulations of cells involved in innate and adaptive immune responses, such as dendritic cells, monocytes/macrophages, and NK cells, while B and T lymphocytes contribute to vascular and kidney injury in hypertension. CD8+, CD4+ T-helper cells, and B-lymphocytes promote hypertension through NADPH oxidase, ROS, cytokines such as TNF-α and IL-17, and IgG production [28]. Lymphocytes and macrophages promote oxidative stress and renal Ang II production and are involved in leukocyte infiltration, tissue damage, and excess sodium reabsorption in the proximal tubule. ACE1 inhibition influences sympathetic neurotransmission via reduced Ang II production and the enhancement of bradykinin and prostaglandin. Ang II has mild platelet-activating effects in vivo and enhances coagulation [31]. Activation of the sympathetic nerve system increases the inflammation of blood vessels and kidneys, causing hypertension. Regulatory T cells play an important role in hypertension by regulating and reducing the response of the adaptive immune system and helping to reduce the hypertension [32].

Ang II is deeply involved in monocyte production, and monocytes themselves express the Ang II receptor. In mice, macrophages accumulate in blood vessels during hypertension, and Ang II–dependent hypertension is associated with monocyte and M2 macrophage accumulation in the aorta [33]. Ang II activates the AT1R and is involved in the generation of the proinflammatory M1 macrophage phenotype. Ang II enhances Toll-like receptor 4 signaling; stimulates the release of ROS, cytokines, and chemokines from M1 cells; and causes inflammation more efficiently. Excessive ACE1 activity can be decisive in causing immune-induced injury through tissue damage, while activation of the AT2R promotes M2 differentiation, anti-inflammatory IL-10 release, and tissue remodeling.

Activation by Ang II is also known to be a major determinant of atherosclerosis [34]. When blood vessels are strained due to high blood pressure or diabetes, the endothelial cells of the blood vessels are damaged, and the function of the endothelium to prevent arteriosclerosis is lost. Macrophages are most abundant in atherosclerotic plaques and express the AT1R. Macrophage polarization by the AT1R is also involved in the exacerbation of renal failure by accelerating atherosclerosis. The Ang II induces MCP-1, VCAM-1, and E-selectin expression; endothelial cell migration; and the adhesion of human monocytic cells (U-937) to HUVECs [35]. Ang II plays critical roles in inflammatory diseases and macrophage recruitment as a powerful proinflammatory factor, contributes to the functional skewing of cardiac-infiltrated monocytes/macrophages, and is involved in autoimmune myocarditis development. 

The RAAS and Ang II also have a significant effect on the platelet and coagulation/fibrinolytic system [36]. Ang II caused mild activation of the coagulation cascade with increases in plasma levels of the thrombin–antithrombin complex and prothrombin fragment F1 + 2 shown as biomarkers of thromboembolic risk. Ang II also induces the mRNA expression of PAI-1 in cultured endothelial cells [37]. Platelet activation/aggregation by Ang II, hypercoagulation, coagulation control dysfunction, and fibrinolysis suppression by plasminogen activator inhibitor-1(PAI-1) can occur locally upon SARS-CoV-2 infection, which may result in the formation of plugs in the body [38].

## 4. Possible Involvement of *ACE1* I/D Polymorphism in the Aggravation of COVID-19 Symptoms

### 4.1. Literature and Database Searches (Epidemiological Studies)

Studies conducted prior to the SARS-CoV-2 pandemic era have already shown an association between the DD genotype and incidence, morbidity, and mortality risk in patients with acute respiratory distress syndrome (ARDS) [39]. In addition, the frequency of the D allele was shown to be significantly higher in the hypoxemic group than in the non-hypoxemic group, whereas there was no significant difference between SARS cases and the control [40].

According to our database analysis, the *ACE1* II genotype frequency in a population was significantly negatively correlated with the number of deaths due to SARS-CoV-2 infection, suggesting that the *ACE1* II genotype may favorably influence the prevalence and clinical outcome of COVID-19 [1]. In addition, a detailed systematic review published by Pabalan et al. shows that the *ACE1* DD genotype may be an important prognostic marker for mortality in Asian COVID-19 patients with acute lung injury / ARDS [41]. Similarly, by conducting an epidemiological study in the population from 26 Asian countries, Pati et al. showed that the *ACE1* D allele is associated with susceptibility to SARS-CoV-2 infection and with the mortality rate [42]. These authors found a significant positive correlation of the D allele of *ACE1* polymorphism with SARS-CoV-2 infection and concluded that the D allele of *ACE1* I/D polymorphism is associated with the rate of infection and mortality. Two other groups have also shown that as the frequency of the *ACE1* II genotype increased, the mortality rate of COVID-19 decreased significantly [43,44]. 

Conversely, however, certain studies have shown that DD genotype is favorable in SARS-CoV-2 infection, or that there is no association between DD genotype frequency and COVID-19 mortality. Delanghe et al. conducted a multiple regression analysis using the data from 25 European countries and showed a negative correlation, i.e., the COVID-19 infection rate decreases as the frequency of the *ACE1* D allele increases [45]. Meanwhile, Cenanovic et al. analyzed the *ACE1* D allele frequencies in 18 selected European populations and compared them with COVID-19 prevalence, mortality, and severity using multivariate linear regression analysis [46]. Consequently, they found no clear statistical evidence to show that the *ACE1* D allele is associated with increased or decreased COVID-19 incidence, mortality, or case severity within the investigated populations.

### 4.2. Studies with Patient Samples (Clinical Studies)

Analysis using patient samples has been practiced in several countries on the Eurasian continent, providing valuable data (Table 1). Calabrese et al. analyzed the association of severe/critical COVID-19 pneumonia with *ACE1* I/D polymorphism in southern Italy [47]. Since the risk of thromboembolism is high, especially in patients with severe COVID-19 pneumonia, and decisive involvement of the balance between ACE1 and ACE2 activities is suggested in the thrombo-inflammatory process, 68 patients with severe COVID-19 pneumonia were divided according to the development of pulmonary embolism (PE) (25 PE + patients, 43 PE- patients). The results showed a statistically significant difference between PE + and PE- patients in assessing *ACE1* I/D polymorphisms (*p* = 0.029). In particular, the prevalence of DD homozygous polymorphisms was significantly higher in PE + COVID-19 patients than in PE- patients (72 vs. 46.5%, respectively; *p* = 0.048). Conversely, heterozygous I/D polymorphisms were significantly lower in PE + patients than in PE- patients. *ACE1* DD homozygosity has been associated with the development of thromboembolism in subjects without a predisposition to other diseases or changes in traditional thrombophilia. In this study, there were no significant differences in serum D-dimer levels between patients with COVID-19 with or without PE. However, the increased C-reactive protein observed in patients with PE suggests the important role of inflammation in the pathogenesis of thrombotic complications in patients affected by SARS-CoV-2 infection. In conclusion, the results of this study suggest that genetic factors are involved in the susceptibility to thromboembolism occurring in COVID-19. *ACE1* DD polymorphisms associated with high levels of both ACE1 and Ang II may represent genetic risk factors. 

In southern Italy, a small study was also conducted using 27 patient samples, focusing on severe respiratory failure associated with COVID-19 pneumonia [48]. At admission, 24 patients suffered from severe respiratory failure with paO2/FiO2 <100 mmHg and were treated with mechanical ventilation. Genetic testing of these patients showed that 19 (73%) of critically ill patients had DD genotype, whereas six (23%) had ID genotype. Only two patients (8%) showed II polymorphism. Studies on patients with ARDS prior to COVID-19 infection have demonstrated an association between DD genotype and cases, morbidity, and mortality risk. 

In Spain, a comparative study was conducted using 204 age-matched controls with 204 COVID-19 patients (137 non-severe and 67 severe ICU cases) [49]. Researchers determined the presence of *ACE1* I/D and *ACE2* rs2285666 polymorphisms and found that severe COVID-19 cases were associated with hypertensive male gender, hypertension, hypercholesterolemia, and the *ACE1* DD genotype (Table 1 shows only the results of males). In multivariate logistic regression analysis, hypertension and male gender remained as independent and significant predictors of severity. The researchers also examined *ACE2* polymorphisms but found no association between them and disease outcomes. Poor outcomes of COVID-19 infection were associated with male gender, hypertension, hypercholesterolemia, and the *ACE1* genotype. The study suggested that *ACE1* I/D polymorphism may affect the severity of COVID-19, but the effect was dependent on the state of hypertension. The D allele was found to be significantly associated with hypoxemia in comparison to patients with non-hypoxemia. However, no association was found between individuals with the DD genotype and those with COVID-19 infection. In conclusion, the study states that the severity of COVID-19 symptoms may depend on age, diabetes, hypertension, and *ACE1* gene polymorphisms. 

In India, Verma et al. genotyped *ACE1* I/D polymorphisms using RT-PCR with a sample of 269 COVID-19 patients and then performed a statistical analysis of the association between COVID-19 symptoms and *ACE1* I / D polymorphism [50]. They found that the frequency of the *ACE1* DD genotype and D allele was significantly higher in patients with severe COVID-19. *ACE1* DD genotype, D allele frequency, older age (≥46 years), and presence of diabetes/hypertension were significantly higher in severe COVID-19 patient. The lack of an *ACE1* I allele was significantly correlated with patients with particularly severe COVID-19. These analyses were performed while adjusting for other factors such as age, gender, marriage status, income, diabetes, and hypertension, but the DD genotype was still significantly associated with clinical outcome of COVID-19, carrying a 3.6-fold higher risk. 

A similar analytical study is being conducted for the first time in Turkey [51]. Clinical data on COVID-19 in the Middle East are invaluable because if the D allele is indeed a risk factor, a big remaining puzzle is why the COVID-19 impact is much lower in the Middle East, where significantly more people have this allele as compared to the people of Europe. According to Gunal et al., age and the frequency of common comorbidities increased significantly in the infected group as the severity increased from asymptomatic as expected. The relationship between the *ACE1* genotype and severity was then analyzed, and they found that *ACE1* II genotype was the predominant genotype (50%) among asymptomatic patients, while the DD genotype was dominant among patients with severe symptoms (63.3%). This study indicates that the *ACE1* II genotype is protective against severe COVID-19.

Most of these findings suggest that heterozygous and non-II homozygous (DD + ID) genotypes of *ACE1* are risk factors for disease aggravation. Conversely, there is a controversial clinical report from the Czech Republic that does not match these data. Hubacek et al. analyzed 408 SARS-CoV-2-positive COVID-19 survivors (163 asymptomatic and 245 symptomatic) using Czech patient samples [52]. The results showed that the frequency of *ACE1* II homozygotes was significantly higher in COVID-19 patients than in the controls. Importantly, this difference was only seen in symptomatic subjects. Thus, they concluded that *ACE1* I/D polymorphism could have the potential to predict the severity of COVID-19, with II homozygotes indicating an increased risk of symptomatic COVID-19. This result appears to be consistent with the epidemiological data published by Delanghe et al. showing an inverse correlation between the prevalence of COVID-19 infection and the frequency of the *ACE1* D allele [45]. 

As described above, among the results using clinical samples, only Hubacek et al. showed the opposite results that II homozygotes, instead of *ACE1* DD, may be associated with an increased risk of symptomatic COVID-19. The reason for these discrepancies is unknown at this time. They added the phrase "at least in Eastern European Caucasians" with that in mind and suggested that differences in geographical and ethnic factors may be behind it. Probably what is important in their study is the fact that Czech patients with genotype II were poor symptomatically, or at least there was no evidence that they were in better condition than those with the DD genotype. Since they analyzed only survivors, it is important to find out what the results would be if the deceased were included. 

Meanwhile, we encountered an interesting article from Lebanon. Using 266 subjects (142 cases and 124 controls), the study found that individuals with the II genotype have a higher risk (OR = 2.373) of contracting COVID-19 as compared to the controls [53]. These results seem to confirm Delanghe’s simulations in patients. Subjects with the DD genotype had a higher probability of experiencing severe COVID-19 symptoms conversely (OR = 7.173), to be hospitalized, and/or to be hypoxic. As to the reason the D allele becomes more resistant to infection, Delanghe et al. proposed a possible association of the D allele with a reduced expression of ACE2 [45]. If this is the case, it will be supported from the virological point of view. However, this can also be a double-edged sword that reduces the beneficial effects of ACE2 on the lungs and other organs. Therefore, the loss of ACE2 can ultimately result in a detrimental enhancement of the Ang II/AT1R response. It will be interesting to investigate the possibility that DD genotype individuals are more resistant to viral infection initially but that the symptoms become more severe once infected as compared to those with *ACE1* II genotype. However, this theory does not well explain the low SARS-CoV-2 infection rate in East Asia, where the frequency of II is much higher than in other areas, such as Europe and the Middle East [1,54].

As to the deleterious effect of *ACE1* I/D polymorphism on COVID-19 symptoms, this situation seems to resemble the relationship between the *ACE1* DD type (D allele) and various comorbidities such as cardiovascular diseases [55]. With regard to comorbidities, most of the database and clinical studies seem to support the involvement of the *ACE1* D allele and DD genotype, while some reports do not support it, and others deny it. However, this result is not surprising given that both comorbidities and COVID-19 are complicated diseases involving many factors in their establishment. In addition to further analysis in different regions and more patient samples, new analyses are expected that consider the involvement of other genes, as described below. 

### 4.3. Possible Involvement of Polymorphic Alu Elements and Their Functional Aspects

Most of the genome is occupied by introns, and the gene exons that encode proteins make up only about 1% of our DNA. The interspersed repeats of mobility in our genome amount to about 45%, but they may be presumed to be non-functional "junk DNA", with the rare exceptions that cause genetic disease. This indicates that only a small part of the function of the entire genome is understood [56]. Structural variations caused by Alu insertion have already been shown to be associated with the risk of many human diseases. The first known disease due to the insertion of a moving element was the insertion of a long scattered element-1 (LINE-1) that interrupted the coding exons of coagulation factor VIII, which is responsible for hemophilia A [57]. Thus far, more than 120 LINE-1 mediated insertions causing hereditary diseases have been reported in humans [58]. 

A growing body of evidence that Alus play an important regulatory role in gene expression in a wide range of physiological processes is accumulating [59]. Recently, based on the hypothesis that a subset of common transposable polymorphisms affects human health, Payer et al. performed genome-wide association studies (GWAS) to identify a number of candidate loci for mutants caused by polymorphic Alu elements. They found 44 Alu insertion polymorphisms showing strong linkage disequilibrium (LD), and single nucleotide polymorphisms (SNPs) were most strongly associated with the disease phenotype [58]. Therefore, it is likely that there is genetic evidence that the Alu mutant is functionally effective at these loci. Given that each insertion produces a structural feature of about 300 bp, the Alu variant is likely to result in causing genetic functional alterations, not just as a bystander. 

Among many Alu elements, the *ACE1* polymorphism is a good example of how much research has been done thus far. Despite the fact that the Alu is an intron insertion, it can affect gene expression through several mechanisms, including both genetic and epigenetic pathways [60]. At the molecular level, it has also been reported that the presence of the Alu element within intron 16 probably affects the promoter activity of *ACE1*, which acts as a transacting repressor for RNA polymerase II activity [61,62]. Strikingly, ACE1 levels in the blood are determined by this I/D polymorphism [63]. Individuals with the DD genotype have been shown to have significantly higher plasma ACE concentrations and ACE activity than people with type II and ID, showing complete LD with Alu DD. 

### 4.4. Possible Interplay between ACE1 and Other Alu Variants

In searching for candidate genes that may act in cooperation with the *ACE1* variants, we have chosen the following criteria: the gene in question (1) has a function that can explain a systemic infection of COVID-19 that includes many organs and systems, (2) is associated with a comorbidity that leads to COVID-19 aggravation, and (3) shows an apparent ethnic difference in its genotype frequency between western and eastern populations. For this, we were particularly interested in the report by Wang et al. [64]. They analyzed the frequency of *ACE1* Alu insertion and *APOBEC3 (A3B)* deletion using a database. A3B proteins are editing enzymes that can interfere with the retrotransposition of endogenous retroelements such as the LINE and Alu. Surprisingly, Wang et al. found the variation curves for *ACE1* insertion and *A3B* deletion among the geographic regions to be almost parallel, both rising continuously along the out-of-Africa expansion route. Alu retrotransposons have undergone many anthropological studies in the context of human evolution. Accordingly, these authors proposed an interesting hypothesis that functional loss of *A3B* provided an opportunity for enhanced human adaptability and survival in response to the environmental and climate challenges arising during the migration from Africa. In this study, they also found that Alu insertion of other genes such as *FXIII* and *CDH13* were increased, and these variants clearly showed ethnic differences in their genotypes [64]. Human A3Bs can also interfere with the replication of exogenous retroviruses such as human immunodeficiency virus (HIV) and hepatitis B virus (HBV) [65,66]. Because intronic Alu is known to regulate gene expression through several mechanisms, both genetically and epigenetically, as described [60], we then focused on the I/D polymorphisms of the *FXIIIB* and *CDH13* genes, expecting the possibility of their interplay with *ACE1* variants. 

FXIII is a transglutaminase and supports platelet adhesion and spreading as well as clot retraction, suggesting that FXIII is important for the stabilization of platelet-fibrin clots. FXIIIB has been studied in recent years for its association with the development of various hemorrhagic disorders, including ischemic stroke, venous thrombosis, coronary artery disease, and myocardial infarction [67,68,69]. Importantly, FXIIIB functions antagonistically with tissue-type plasminogen activator (tPA) in the blood coagulation cascade. Recently, Matsuyama et al. proposed a possible role of the STAT3-PAI-1 signaling node in a chaotic chain reaction underlying COVID-19 pathophysiology. Viral proteins Nsp1 and Orf6, produced upon SARS-CoV-2 infection, efficiently inhibit STAT1 function in the cells. Suppressed STAT1 increases STAT3 activity, and STAT3 eventually upregulates PAI-1. This increased PAI-1 activity may result in IL-6 production through Toll-like receptor 4, which in turn activates more STAT3. Elevated STAT3 also activates PD-L1 in endothelial cells, leading to T-cell lymphopenia [70]. *FXIIIB* polymorphism AluYa5 insertion has been reported to confer a risk of coronary atherosclerosis [61]. Furthermore, intra- and extracellular FXIIIs are indicated to support the immobilization and killing of bacteria as well as phagocytosis by macrophages, strongly suggesting that FXIII may also function in innate immunity [71]. With an analogy to the *ACE1* Alu I/D polymorphism, the *FXIIIB* I/D type can also be involved in the difference in the susceptibility of Westerners and Asians against COVID-19, given the apparent difference in the genotype frequency seen between the those in the west and east.

Genetic variation of the cadherin gene *CDH13* was significantly correlated with certain clinical parameters, such as diastolic blood pressure, triglyceride, adiponectin, and insulin levels [72]. The *CDH13* polymorphism is also associated with a decreased risk of developing hypertension when compared to non-hypertensive individuals [73]. A reduction in plasma T-cadherin levels is associated with increasing severity of coronary artery disease and a higher risk of acute coronary syndrome [74]. Alu insertions of this gene, along with *ACE1*, *FXIIIB*, etc., are often studied to provide valuable information for ancestrality and genetic differentiation [64]. However, the relationship between CDH13 and COVID-19 symptoms is not sufficiently studied at present. 

Li et al. also investigated known Alu polymorphisms of genes such as *progesterone receptor (PGR)* and *plasminogen activator (PLAT)* in addition to the *ACE1* and *FXIIIB* genes for their biological function, role in disease, and effect on COVID-19 [60]. Of particular interest are the variations and functional changes in the *PGR* gene, since this gene has already been shown to have reduced transcript and protein stability and reduced receptor responsiveness to progestin due to Alu intron insertion [75]. This functional relationship of Alu with intron insertion is similar to the results seen with that of ACE1. However, the frequency of Alu insertion of the *PGR* gene is significantly higher in Europeans than in the Asian population, which is the opposite of Alu polymorphisms in *ACE1*, *FXIIIB*, and *PLAT*. Therefore, this suggests that the A3B rules do not apply to all Alu polymorphisms. Nevertheless, the effect of COVID-19 on clinical symptoms by Alu’s variant, which includes *PGR* in addition to *ACE1* and *FXIIIB*, may be worth further study. There is also a significant interaction between *PAI* gene polymorphisms, *PAI4G5G* and *ACE1* I/D polymorphisms, at plasma tPA and PAI-1 levels [76]. These results support the notion that interactions among renin–angiotensin, bradykinin, and the fibrinolytic system may play important roles in tPA and PAI-1 biology. Furthermore, it is noteworthy that Ang IV synthesis in the RAAS regulates the release of PAI-1 from endothelial cells when Ang II activation results in an imbalance of ACE1/ACE2 [77]. *PLAT* encodes tissue tPA found in endothelial cells that promotes fibrinolysis, exerting its effect by converting plasminogen to plasmin. Plasmin breaks down fibrin polymers, or blood clots, into D-dimer, among other breakdown products. The activity of tPA is inhibited by PAI, which is secreted by endothelial cells, among other cell types and tissues. However, the frequency of the Alu insertion of this gene does not seem to be different between Europeans and Asians. Nevertheless, given the importance of the coagulation/fibrinolysis system in the pathogenesis of COVID-19, the function of the Alu variant of the *PLAT* gene cannot be ignored in future studies. tPA is currently being attempted as a therapeutic regimen for various cardiovascular diseases and ARDS associated with COVID-19 [78].

As described, the relationship between comorbidities and *ACE1* Alu I/D polymorphism has been extensively studied, and there are many reports supporting a positive relationship between their pathophysiology and the DD genotype. It has also been shown that it may be at risk for diseases with higher or lower associations between different genes rather than a single gene [79]. Dai et al. showed that high levels of ACE1, kallikrein1 (KLK1), and IL-6 were detected in acute myocardial infarction (AMI) patients with the D allele [80]. Simultaneous increases in Ang II and KLK1 serum levels significantly increased the risk of AMI. Individuals with the *ACE1* DD and *KLK1* GG genotype combination showed a significantly increased risk of AMI compared to individuals with the *ACE1* II and *KLK1* AA genotypes. Although the *KLK1* AA genotype is independent of the Alu, possible interplay of the *ACE1* Alu polymorphism with another gene polymorphism is noteworthy.

### 4.5. ACE1 I/D Genotype Likely to Match Other Alu Gene Polymorphisms at the Individual Level

Wang et al. showed that the Alu I/D genotypes of certain genes, such as *FXIII* and *CDH13,* present geographic distributions analogous to those of *ACE1* [64]. This result indicates that people of the *ACE1* II (or DD) type may have II (DD) more frequently than other Alu I/D genotypes, at least at the population level. To see this overlap at an individual level, a comprehensive analysis using cells derived from 17 Japanese and 22 Caucasoid individuals (including two Africans) was conducted. Alu I/D polymorphisms of *ACE1*, *FXIIIB*, and *CDH13* were determined using cancer-derived cell lines established from both Western and Japanese individuals. Figure 1 shows at a glance, the I/D genotypes of these three genes were overwhelmingly II or ID in Japanese individuals, whereas there were many DD or ID types present among Europeans. This strongly suggests that the genotypes of these variants are likely to match and that the effect when they are combined may not be negligible. This result does not seem surprising from country-specific population data, but it is important to show this fact at the individual level. Based on these data, future analysis in combination with *ACE1* and other gene polymorphisms using actual COVID-19 patient samples is of interest.

Although this review has focused primarily on other Alu gene polymorphisms that may interact with *ACE1* polymorphisms, especially with regard to COVID-19 exacerbations, it should be emphasized that the candidates for susceptibility genes are not limited to these gene genotypes. In addition to the genes examined here, various genes and their variants, such as alfa*1-anti-trypsin (AAT)* genotypes, *ABC transporters*, and *MHC*, have been studied as factors that may determine the pathogenesis of COVID-19 [18].

## 5. Discussion

The subject of this review is that there are ethnic differences in COVID-19 symptoms and susceptibility to infection, and the reasons for this were discussed with special reference to *ACE1* I / D polymorphism. Although other factors such as the emergence of Delta strains and the effects of vaccination have been involved recently, including now (as of August 2021), the fact that the people in Western countries are overwhelmingly affected by this virus remains unchanged. However, since epidemiological information changes from moment to moment, it is necessary to carefully monitor future changes in epidemiological trends. It is true that the infection of mutant strains is rapidly spreading not only in Western countries but also in Asian countries. However, according to previous virological experience, especially that of RNA viruses including coronavirus, the emergence of new mutants was, of course, expected. This is because, given a condition in which viruses are likely to replicate rather easily, the type of virus that is appropriate for the environment, more susceptible to spread, and more aggressive is selected. That is exactly what the world’s social environment is today, and as the rapid expansion of Delta strains around the world shows, it is a favorable environment for viruses. However, as one of the survival strategies, it is well known that the virus is attenuated in a state where it is difficult for the virus to spread. Thus, the best way to control this pandemic is for each person to understand how the virus infection is less likely to spread, to implement it, and international vaccination should be performed as widespread as possible.

It is well known that there are differences in susceptibility and resistance to viruses between populations. For example, Africans are more resistant than Europeans to yellow fever and malaria. Measles is more likely to be severe in people from the Pacific and Africa. Therefore, it is not strange that there is a clear ethnic difference in the susceptibility and resistance to COVID-19, especially in the number of deaths between Westerners and East Asians. We have investigated the effects of *ACE1* gene I/D polymorphism on SARS-CoV-2 infections and have shown that DD and the D allele may be risk factors for COVID-19 aggravation. However, in reality, the difference in the frequency of the D allele between Westerners and East Asian people is not large (for example, according to our analysis, about 57% of Caucasians have a D allele, but about 32% of Japanese also have this allele). To solve this puzzle, we searched for genes and their polymorphisms that may operate in concert with the *ACE1* I/D polymorphism. It was shown that *ACE1* II carriers frequently have the same II genotype as the other two Alu-related genes, *FXIIIB* and *CDH13*. This suggests that even if the *ACE1* I/D polymorphism alone shows only a small difference, having the I/D genotype of other genes at the same time could make a big difference in COVID-19 symptoms between the west and the east. Therefore, it is worth considering the possibility that certain gene combinations are involved in the development of severe COVID-19, and these may determine changes in pathogenically important quantitative traits. However, the association between *ACE1* I/D polymorphism and COVID-19 aggravation may be due to a linkage disequilibrium to closely mapped genes, other than Alu-related genes, that have not yet been identified. It is not negligible that there are non-random correlations between alleles or genetic markers (polymorphisms) at multiple loci in a population of organisms, and the frequency of their particular combination (haplotype) may be significantly higher. Genome-wide studies have already been conducted in several countries for SARS-COV-2 infection [81]. Use of genome-wide association studies (GWAS) may suggest partial roles of several gene clusters and genetic liability for the development of COVID-19 in the near future. Infectious diseases, including COVID-19, are originally polygenic diseases. 

As each human has a different genetic structure, it is not easy to investigate the genetic factors for disease. However, this is possible with inbred animals such as mice. In the mouse hepatitis virus (MHV), a type of coronavirus, the mode of inheritance of susceptibility/resistance to mouse hepatitis strain 3 (MHV-3) was determined by typing the set of *AXB/BXA* recombinant inbred (RI) strains derived from resistant A/J (A) and susceptible C57BL/6J (B) progenitors for susceptibility to infection as determined by the severity of liver pathology [82]. Similarly, resistance to certain flaviviruses is under the control of a single gene, with the resistance gene predominant. Macrophages in susceptible mice support viral growth well, but viruses have also been found to grow poorly in resistant animals [82]. The virus infects and propagates in the host cell and repeatedly infects the target cells, one after another, causing changes in organs or tissues. For that purpose, it is necessary for the cells to have an appropriate receptor such that the virus bound to the cells invades and produces a progeny virus at an appropriate site. Cellular factors for that purpose are also diverse, depending on the virus, and many have been identified. Conversely, the main reasons for resistance are the lack of any of these host factors. In addition, the action of antiviral factors is important, of which interferon is the most studied. Many other common or virus-specific host factors that act on the innate immune system are also known.

Since susceptibility and resistance to disease are related not only to genetic differences but also to environmental and social factors, the differences cannot be considered based on genetic factors alone. In addition to genes, there are several possible factors that may explain the apparent difference in the number of COVID-19 deaths between Western and East Asian societies. As SARS-CoV-2 spreads through airborne infection, this pandemic has significant implications for the cultural and social determinants of health and illness. Sociobehavioral differences between Eastern and Western countries are clear, e.g., the use of masks and habits such as kissing, hugging, and shaking hands. Furthermore, in East Asia, the agricultural society was established thousands of years earlier than in Europe, enabling explosive population growth, and this made it possible for various respiratory and gastrointestinal infections, which were originally derived from other animals, to become established in human society. During that time, many mutant viruses appeared in these areas, and thus it is not difficult to imagine that East Asians are experienced in exposure to highly infectious and more pathogenic viruses as compared to Europeans. There is no doubt that East Asians have been infected with various viruses earlier than Europeans and have been under selective pressure. Since most studies included in this review are correlational studies, there is a risk of confounding by other factors such as geopolitical and socioeconomic characteristics of each country. For example, Gelfand et al. conducted a global analysis on the relationship between cultural tightness–looseness and COVID-19 cases and deaths. As a result, it is suggested that tightening social norms may bring tremendous advantages in times of collective threat [83]. Therefore, disease susceptibility should also be considered from such sociobehavioral and hygienic aspects of people [84]. 

Since leaving Africa about 100,000 years ago, people have spread to the continents of Eurasia, Sahul, and the Americas and have undergone various genetic changes. This has clearly brought about a change in appearance, but it also caused significant changes inside the body. These changes must have been beneficial for people in migration, to adapt to the climate and environment of the land they reached and to survive against infectious diseases. Otherwise, it is unlikely that such specific genetic alterations or subtypes would become dominant and be fixed in a particular population.

## Figures and Tables

**Figure 1 genes-12-01572-f001:**
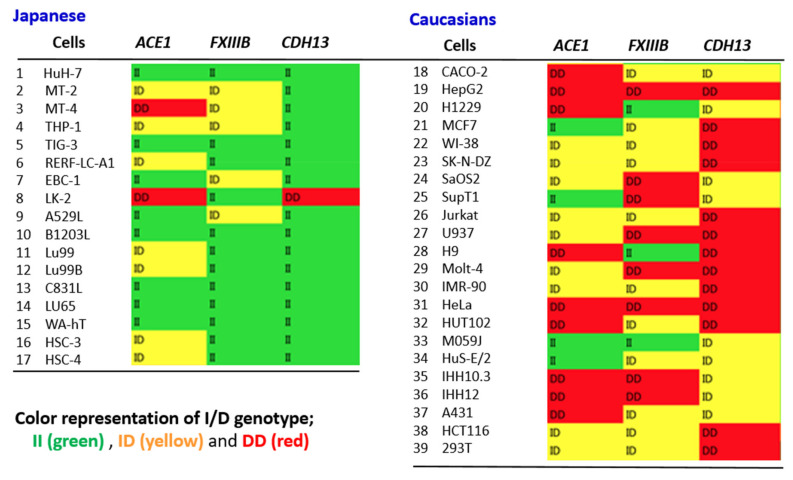
Alu I / D polymorphisms of *ACE1*, *FXIIIB* and *CDH13* in cells established from Japanese and Caucasian individuals. The Caucasian cell lines also include two African-derived cells, HeLa and HUT-102.

**Table 1 genes-12-01572-t001:** The association of COVID-19 symptoms with *ACE1* I/D polymorphism using patient specimens.

Country	Reference	*ACE1* I/D Genotype	Control	COVID-19 Patients					Statistics
Italy	Calabrese, 2020 [47]		Control (*n* = 111)			PE- (*n* = 43)	PE+ (*n* = 25)		*p*
		II	13 (11.7%)			2 (4.7%)	3 (12%)		
		ID	50 (45%)			21 (48.8%)	4 (16%)		
		DD	48 (43.3%)			20 (46.5%)	18 (72%)		0.03
Italy	Annunziata, 2020 [48]			Patients (*n* = 27)		PaO_2_/FiO_2_ (mmHg)			
		II		2 (8%)		>200			
		ID		6 (23%)		86.9 ± 15.3			
		DD		19 (73%)		75.6 ± 11.3			
Spain	Gomez, 2020^1^ [49]		Control (*n* = 248)	COVID (*n* = 125)					*p*
		II	40 (16%)	6 (5%)					
		ID	123 (50%)	66 (53%)					
		DD	85 (34%)	53 (42%)					0.13
					Mild (*n* = 72)	Severe (*n* = 53)			*p*
		II			4(5%)	2 (4%)			
		ID			43(60%)	23 (43%)			
		DD			25(35%)	28 (53%)			0.04
India	Verma, 2021 [50]				Mild (*n* = 149)	Severe (*n* = 120)			OR (*p*-value)
		II			74 (49.7%)	42 (35.0%)			1 (ref)
		ID			58 (38.9%)	48 (40.0%)			1.54 (*p* = 0.17)
		DD			17 (11.4%)	30 (25.0%)			3.69 (*p* = 0.002)
Turkey	Gunal, 2021 [51]			Asymptomatic (*n* = 30)	Mild (*n* = 30)	Severe (*n* = 30)			*p*
		II		15 (50%)	7 (23.3%)	9 (30%)			0.08
		ID		4 (13.3%)	8 (26.7%)	2 (6.7%)			0.09
		DD		11 (36.7%)	15 (50%)	19 (63.3%)			0.12
Czech Republic	Hubacek, 2021 [52]		Control (*n* = 2579)	Asymptomatic (*n* = 163)	Symptomatic (*n* = 245)			OR Cont. vs. Asympt.(*p*-value)	OR Cont. vs. Sympt.(*p*-value)
		II	547 (21.2%)	36 (22.1%)	71 (29.0%)			1.15 (*p* = 0.55)	1.78 (*p* = 0.002)
		ID	1331 (51.6%)	87 (53.4%)	123 (50.2%)			1.14 (*p* = 0.49)	1.27 (*p* = 0.16)
		DD	701 (27.2%)	40 (24.5%)	51 (20.8%)			1 (ref)	1 (ref)

^1^ Only results of males were provided in the original paper, which demonstrated an association of an adverse outcome of COVID-19 with male gender, hypertension, hypercholesterolemia and the *ACE1* genotype.

## Data Availability

Data was either obtained from the individual studies cited within the article or contained within the article.

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
