# Peer review of "Angiotensin–Converting Enzyme (ACE) 1 Gene Polymorphism and Phenotypic Expression of COVID-19 Symptoms"

_genes, 2021, doi:10.3390/genes12101572_

Round 1
Reviewer 1 Report
Page 2: SARS-CoV-2 enters host cells through the interaction of its spike protein with the entry receptor, ACE2 (6), and its expression levels are most likely to be associated with susceptibility to SARS-CoV-2.
Provide references for this, studies reporting association between expression and susceptibility?.
Accordingly, many studies targeting ACE2 have been conducted, including the development of therapeutics.
Same, reference some of the studies.
The binding of SARS-CoV-2 to ACE2 and processing of the S protein by TMPRSS2 may result in the downregulation and depletion of ACE2.
Studies reporting this effect?.
The studies reporting association between ACE I/D and COVID-19 are based on very small number of patients, and they lack statistical power. This makes this revision of limited interest in reference to the association between COVID and ACE I/D. In these studies is almost impossible to perform a multivariate analysis including hypertension and other covariables, due to the very low numbers.
Also, the authors reviewed the studies that analysed the correlation between frequency of the variants and disease prevalence or mortality. These studies are of limited interest because the data of prevalence and mortality are biased by multiple factors, including the number of tests, the political and sociological characteristics of each country, etc. All these should be discussed. The authors could consider this ms:
Michele J Gelfand et al. The relationship between cultural tightness–looseness and COVID-19 cases and deaths: a global analysis. www.thelancet.com/planetary-health Published online January 29, 2021 https://doi.org/10.1016/S2542-5196(20)30301-6
Reviewer 2 Report
very complex review can I have statistical help for table 1,were contradictory results are shown and only males were studied,what about females data?There contradictory results may in certain geographic areas,such as Japan and Caucasians, especially in subgroups of patients.to speed up the process a statistician should look at this with a meta-analysis and verify male/female.
Reviewer 3 Report
The introduction needs to be extensively re-organized. The transition from ACE2 to RAAS was extremely abrupt and without much segway. How does COV2 disrupt RAAS signaling? Please provide explanation. Several proteins are abbreviated without once mentioning full names e.g. Ang I and Ang II. The manuscript is very confusing and lacks a flow and definition. Title of the manuscript gives an impression that the AC I polymorphism would be the main subject of discussion. If that is the case, there is a LOT of other subtopics discussed that dont need to be so extensively analyzed. Figure 1 is wrongly placed and describes details that arent fully described through the manuscript.
Round 2
Reviewer 1 Report
I have revised the ms and i feel it was revised accordingly to my commentaries-suggestions.
Reviewer 2 Report
No further comments, the meta-analysis might be the next future development.